# Factors hindering integration of care for non-communicable diseases within HIV care services in Dar es Salaam, Tanzania: The perspectives of health workers and people living with HIV

**Tausi Haruna**[1]*, **Magreat Somba**[2], **Hellen Siril**[2,3], **Gladys Mahiti**[2], **Francis August**[2], **Anna Minja**[2], **David Urassa**[4], **Edith Tarimo**[5], **Ferdinand Mugusi**[6]

1 Department of Fundamentals of Nursing and Basic Sciences, Hubert Kairuki Memorial University, Dar es Salaam, Tanzania, 2 Department of Development Studies, Muhimbili University of Health and Allied Sciences, Dar es Salaam, Tanzania, 3 Department of Health Care Quality Improvement and Public Health Evaluations, Management and Development for Health, Dar es Salaam, Tanzania, 4 Department of Community Health, Muhimbili University of Health and Allied Sciences, Dar es Salaam, Tanzania, 5 Department of Nursing Management, Muhimbili University of Health and Allied Sciences, Dar es Salaam, Tanzania, 6 Department of Internal Medicine, Muhimbili University of Health and Allied Sciences, Dar es Salaam, Tanzania

* tausiharuna2015@gmail.com

**Data Availability Statement:** All relevant data are within the paper and its Supporting Information files.

## Abstract

### Background

Global mortality attributable to non-communicable diseases (NCDs) occurs in more than 36 million people annually with 80% of these deaths occurring in resource limited countries. Among people living with HIV and AIDS (PLHA) studies have reported higher prevalence's of NCDs compared to the general population but most studies do report a narrow range of NCDs commonly hypertension, diabetes and neoplasms and not all. In addition, there is limited reporting, integration of systematic screening and treatment for all NCDs among PLHA attending care, suggesting the NCD burden among PLHA is likely an underestimate. Little is known about factors facilitating or hindering integration of the care and treatment of NCDs within HIV care and treatment clinics (CTCs) in Tanzania.

### Objective

To explore the perceptions of PLHA and health workers on factors facilitate or hinder the recognition and integration of care for NCDs within CTCs in Dar es Salaam.

### Methods

Inductive content analysis of transcripts from 41 in-depth interviews were conducted with 5 CTC managers (CTC Managers), 9 healthcare providers (DHCP) and 27 people living with HIV (PLHA) attending CTCs and with co-morbid NCDs.

**Funding:** H.S; D43TW009775 Muhimbili University of Health and Allied Science in collaboration with Fogarty International Centre of the National Institutes of Health (NIH): www.nih.gov T.H; U91HA06801; AfyaBora Project; United State of America President's Emergency Plan for AIDS Relief (PEPFAR) and Health Resources and Services Administration (HRSA); www.afyaboraconsortium.org. The funders had no role in study design, data collection and analysis, decision to publish, or preparation of the manuscript.

**Competing interests:** The authors have declared that no competing interests exist.

## Results

Four themes emerged; the current situation of services available for care and treatment of NCDs among PLHA in CTCs, experiences of PLHA with co-morbid NCDs with access to care and treatment services for NCDs, facilitators of integrating care and treatment of NCDs within CTCs and perceived barriers for accessing and integration of care and treatment of NCDs within CTCs.

## Conclusions

There was a positive attitude among PLHA and healthcare workers towards integration of NCD services within CTC services. This was enhanced by perceived benefits inherent to the services. Factors hindering integration of NCD care and services included; limited and inconsistent supplies such as screening equipment, medications; insufficient awareness of NCDs within PLHA; lack of adequate training of healthcare workers on management of NCD and treatment costs and payment systems.

## Introduction

Globally, non-communicable diseases (NCDs) kill more than 36 million people each year with 80% of these deaths occurring in resource limited countries [1]. Data published in the Global Burden of Disease 2010 [2], shows that the NCDs are becoming a significant cause of poor health globally, including in resource limited countries, except sub-Saharan Africa, where NCDs are second to the disease burden of HIV/AIDS. A large proportion of reported NCD deaths and chronic illnesses are reported to be caused by cardiovascular diseases, cancer, chronic respiratory diseases, and metabolic and digestive disorders such as diabetes [3].

Among people living with HIV and AIDS (PLHA) in LMICs, the prevalence of NCDs is higher (range 29%-44%) compared to that in the general population (15%-25%) [2]. However, limited reporting and integration of systematic screening and treatment for NCDs using standard guidelines at primary care levels of health services occurs in sub-Saharan Africa [3]; the NCD burden is thus likely to be underestimated among PLHA. In health care settings in Kenya the prevalence of NCDs among PLHA is reported as 11.5% with hypertension being the most common; however the diagnosis and recording of identified persons with hypertension in patients' records was as low as 4.9% [3]. A systematic review and meta-analysis of the prevalence of cardiovascular disease, cervical cancer, depression, and diabetes in PLHA in low-and middle-income countries (LMIC) focusing on sub-Saharan Africa, showed estimates (or range) of 21.1%; 1.3%-1.7%; 24.4% and 1.3–1.8% respectively. The estimated rates for risk factors of NCD such as low high-density lipoproteins (52.3%), hypertriglyceridemia (27.2%), obesity 7.8% and elevated low-density (23.2%) were also reported [4]. Furthermore, the prevalence of the less frequently diagnosed neurocognitive impairment (NCI) among PLWHA in Sub Saharan Africa (SSA) may be higher than expected; a systematic review shows prevalence of NCIs among PLHA from seven SSA countries was 6.5 times higher in PLHA compared to in persons without HIV [5]. The same study showed reduction in NCI prevalence, before (43%), compared to six months after (30%) initiation of anti-retroviral therapy (ART) initiation [5]. These evidences suggest an importance of screening and managing NCDs within care and treatment services provided for PLHA.

Over the past several decades, improvements in health care have led to a decrease in mortality in both the general population and persons living with HIV and AIDS [6], and an increase

in life expectancy in resource limited countries [7]. While infectious diseases, such as HIV/ AIDS and tuberculosis, are particular health burdens for resource limited countries in sub-Saharan Africa [5]; very limited data are available on the epidemiology, natural history and pathogenesis of co-morbid NCDs in the contexts of care for these infectious diseases.

Similar to other LMIC, a very limited number of studies have explored the burden of NCD among PLHA in Tanzania. Amongst PLHA in urban Tanzania the prevalence of hypertension (high blood pressure, defined as a blood pressure of $\geq$ 140/90mmHg, and diabetes (defined according to WHO criteria) was 26%and 4.2% respectively [8].

Furthermore, with regard to NCDs burden it is important to consider factors that facilitate or hinder integration of HIV care services within CTC. A literature review conducted by Haregu et al. looking into policies and models for NCD integration within HIV clinics revealed evidence related to diseases progress, clinical, and management issues [7]. In this study the author identified models of integration centered to Problem, People, Process and Patient [7]. Another study conducted in African countries identified a limited model in Tanzania for integration in the Reproductive Child Health Clinics among women who present for cervical cancer screening. In this Model 1 represented NCD services integrated into centers originally providing HIV care where by women who tested positive were referred to CTC [9].

This study will be the first in Tanzania to describe from the perspectives of PLHA and health care providers the perceptions on factors facilitate or hinder the recognition and integration of care for NCDs within CTCs in Dar es Salaam.

## Methods

### Observations

A 5 days an announced onsite check-list guided observation (1 day per site) at the 5 CTCs was conducted by MDH CQI nurses from different districts. The observation was done for the 1st 20–40 patients who arrived at CTCs for routine care.

A total of 130 PLHA were observed for the 5 CTCs. The nurses observed their NCD screening services they received during the visit as well as checked on the pharmacy for the availability of NCD medications. Each site was observed by a CQI nurse from a different district nurse.

**Study design.** We conducted a descriptive qualitative study which included in-depth interviews and onsite checklist guided observation, in order to obtain a better understanding of the perspectives of PLHA and health care workers on the integration of care for non-communicable diseases within HIV care services. We used purposive sampling to select experienced CTC direct health care providers (DHCP), CTC managers and PLHA who were documented to have a co-morbid NCD, who would be most knowledgeable about NCD services in CTCs and be able to contribute key information to inform our study objectives. We selected and interviewed 27 PLHA, nine DHCP, and five CTC managers.

**Data collection tools.** Semi-structured interview guides were developed to collect information from PLHA, DHCP and CTC manager participants. We prepared In-depth interview guides based on current literature on the recognition and integration of care for NCDs within CTC. The guides comprised of open-ended questions and probes were developed by the first author and shared to research assistants for increasing the depth of the information provided. The inputs given were incorporated for improvement. Piloting of the tools were done among other PLHA not included in this study to explore their experiences with NCDs services (screening, diagnosis treatment) and factors facilitate or hinder integration of care within CTC. This gave us insight on the truly perspectives on NCD integrations and so helped to formulate more relevant questions. The guides also probed for capacity of health care workers to care for PLHA experience when providing NCD services for PLHA within the context of

CTCs and sought recommendations for improving the integration of NCDs services within CTCs. Interview guides were modified depending on the new information emerged during data collection and analysis. Observations were also conducted in each of the five study sites by experienced and trained health care quality improvement nurses not linked to the CTC. The observation check list collected information on NCD screening tests done in routine CTC visit, and availability of/stock of NCD drugs at CTCs.

**Study setting.** The study was conducted in 5 large CTCs located within the five district referral hospitals in Dar es Salaam region in Tanzania. The selected CTCs served a district wide catchment, and serves a large population of PLHA defined as providing services for 120 to 200 PLHA per day. The site selections allowed for attaining the desired sample within the study time, as well as inclusion of a diverse sample of study participants from across all the five districts of the Dar es Salaam region. We recruited adults PLHA with co-existing NCD (aged 18 years and above) who were enrolled and receiving ART services and health care workers (DHCP) who are directly involved in providing routine HIV care and treatment services at CTCs including nurses and doctors and who had been working in CTC for a minimum of 3 months.

This study excluded PLHA aged less than 18 years, too ill to participate in the IDI, newly employed CTC managers and DHCP. Socio-demographic information including, age and gender, were collected from all types of informants, while for DHCPs and CTC managers, information was also collected on their qualifications, job titles, and duration of providing health care services since graduating as health professionals.

**Data collection procedures.** Data collection was done with six (6) research assistants who received a refresh training on qualitative data collection. The research assistants had a mix of pre-graduate diplomas in health and social science para professional field (n = 3) postgraduate degree in Public health (n = 1) and undergraduate degrees in the social sciences or Law (n = 2) fields. All RAs also were up to date on research ethics certification (FHI360 certification course). All RAs had prior experiences collecting qualitative data. For this study the training of RAs was overseen by (HST, SH, SM. and MG). The gender distribution of RAs was five female and one male. A saturation point was reached when a total of forty-one (41) interviews were conducted. Twenty-seven (27) interviews with PLHA, 9 DHCP and 5 CTC managers which lasted about 50, 40 and 35 minutes respectively. None of the RAs had contact with the participants before conducting the first interview. Permission to audio-record interviews was requested and all participants agreed to interviews being recorded. All interviews were conducted within the facilities and measures taken to ensure privacy and that informed written consent was provided by all study participants. Upon completion of each field data collection session for both the in-depth interviews and the observations, narrative data transcription was done on the same day sent to the central study team which included the study PI and co-investigators. All audio-records of the data were uploaded to password protected computers to which the PI and co-investigator's had access, and this provided an additional layer of quality assurance that transcripts were reflective verbatim of each interview session. Feedback sessions with interviewers also discussed probing skills for gaining greater understanding of the meaning behind participant responses to ensure improvements in depth of information provided in subsequent interviews. All interviews were conducted in Swahili.

**Data analysis.** Data coding was done using the NVIVO software version 8 [10]. Coding refers to the breaking down of narrative data into small units of words called codes [11]. We analyzed the qualitative data based on a content thematic approach [12] where by, firstly the transcripts were read several times to obtain a sense of a whole and identifying the meaning units. That means the short sections of the transcripts seems to be meaningful and that relate to our research questions and objectives. Secondly the formulated meaning units were then

**Table 1. Example of the analysis process of the main theme, categories and illustrative quotes from respondents.**

| Theme | Category | Illustrative quotes from respondents |
|---|---|---|
| Current situation of availability care and treatment services for NCDs among PLHA in CTCs | Referrals to NCD clinics outside the CTC | *"We receive many patients with hypertension here at the CTC, about 30 patients per week or roughly 7–8 patients a day but we don't provide care for NCDs here* (at the CTC)*" (DHCP F, 41 years).* |
| | Writing prescriptions for NCD medications at the CTC for patients to purchase | |
| Experiences of PLHA with co-morbid NCDs with access to care and treatment services for NCDs | Access to care for NCDs within the same hospital where a CTC is located | *At the CTC clinic they said I have to be treated at a different place, not the CTC but at another NCD clinic in the hospital or at any other hospital, because at the CTC they only deal with the ARVs (Patient, F, 52 years)* |

condensed in short summary forms as indicated by Graneheim and Lundman [12] as condensed meaning units. Thirdly the condensed meaning units, codes were abstracted further and developed codes and preliminary groups, categories were developed. Then categories which refer high level of abstraction and still indicate the manifest of transcripts. Lastly the constant comparison between categories and the rest of the material themes were constructed and reflected the latent content of the text (Graneheim and Lundman [12]). Coding was conducted with the Kiswahili transcripts to remain close to the text. Codes were then translated into English to develop categories and themes. All the researchers are native Swahili speakers. See Table 1 for more clarification.

## Trustworthiness

There are numbers of criteria used in evaluating trustworthiness of the qualitative papers, namely credibility, transferability, dependability and confirmability [13]. In this paper credibility was ensured by purposively sampling in selecting the study participants who are eligible for participation. These participants were key to our research question "What are the factors that facilitate or limit integration of care for Non-Communicable diseases within HIV care services?" The use of observation checklists, IDs and the use of PHLA, DHCP and CTC managers have enhanced credibility of the study.

## Ethical considerations

Ethical approval to conduct this study was granted by the National Institute of Medical Research (NIMR) (reference numbers; NIMR/HQ/R.8A/Vol.IX/3362). Permission to conduct the study was requested from hospital and CTC authorities, each group of participants had their own written informed consent. Participants also consented for their interviews to be recorded and anonymity was observed in all data collection tools.

## Results

### Onsite checklist-based observation findings

A total of 130 PLHA were observed from the 5 CTCs. Each site was observed by a continuous quality improvement (CQI) nurse from a different district. Weight is the most commonly measured parameter for all the PLHA during routine CTC visits; and is assessed in all patients regardless of the type of visit (new patients or follow up patient). However, none of CTCs assessed height consistently or estimated the BMI for PLHA observed. Heights are taken for patients who are seen for the first time only and none used the height results to calculate and

estimate BMI during our observation. None of the CTCs conducted clinic based random blood sugar tests during our observation. Only one out of the five hospitals routinely measured patients of blood pressures but only when the patient was suspected by the clinician to have an elevated blood pressure.

## Sociodemographic characteristics of the study participants

A total of 27 PLHA with co-morbid NCDs participated in the in-depth interviews. Their mean age was 54 years (ranging from 39–70 years). Twelve (44.4%) were married, 15 (55.5%) were single, widowed or divorced. Five (19%) PLHAs reported no formal education, 17 (62.9%) had completed primary education, two (7%) had completed four years of secondary education while one (4%) had college education. Almost two thirds (59%) reported receiving NCD care at health facilities that where external to where they received HIV CTC services. See Table 2;

**Socio demographic characteristics of DHCP and CTC managers.** The mean age of CTC managers was 43.5 years, 80% were females and all of them had worked in the health sector for an average of 6.4 years (range 2–11 years). The direct health care provides (DHCP) were aged between 25 and 54 years with a mean age of 38.1 years and they all had been working in health sector for an average of 6.4 years (range 3–16 years). Four (44%) were medical doctors (MDs) while 5 (56%) of the DHCP were COs. See Table 3;

**Table 2. Demographic characteristics of people living with HIV/AIDS (PLHA) and NCD comorbidity accessing HIV care and treatment clinic services in 2020, Dar es Salaam, Tanzania (N = 27).**

| Characteristics | Participant N = 27 (%) |
|---|---|
| Age (Years*) | |
| Mean | 54 |
| Range | 39–70 |
| Gender | |
| Male | 4(14.8%) |
| Female | 23(85.2%) |
| Marital status | |
| Single | 3(11.1%) |
| Married | 12(44.4%) |
| Divorced | 3(11.1%) |
| Widow/widower | 9(33.3%) |
| Education | |
| No formal education | 5(18.5%) |
| Incomplete primary education | 2(7.4%) |
| Completed primary education | 17(62.9%) |
| Secondary education (form four) | 2(7.4%) |
| College | 1(3.7%) |
| Occupation | |
| Business | 16(59.3%) |
| Employed | 1(3.7%) |
| Unemployed | 10(37.0%) |
| NCD type | |
| Hypertension | 13(48.1%) |
| Diabetes | 4(14.8%) |
| Hypertension & Diabetes | 7(25.9%) |
| Hypertension & Heart problems | 2(7.4%) |
| Hypertension & All types of ulcers | 1(3.7%) |

**Table 3. Socio- demographic characteristics of CTC managers and direct health care providers (DHCP) in HIV CTCs in 2020, Dar es Salaam, Tanzania (N = 14).**

| Category | Age | Sex | Qualification | Duration of work in health sector |
|---|---|---|---|---|
| **CTC Managers (n = 5)** | | | | |
| Participant 1 | 50 | Male | AMO* | 8 years |
| Participant 2 | 51 | Female | AMO | 6 years |
| Participant 3 | 37 | Female | MD** | 11 years |
| Participant 4 | 36 | Female | MD | 5 years |
| Participant 5 | 34 | Female | MD | 2 years |
| **CTC Health Care Workers (n = 9)** | | | | |
| Participant 1 | 25 | Female | CO*** | 4 years |
| Participant 2 | 28 | Female | CO | 3 years |
| Participant 3 | 36 | Male | MD | 3 years |
| Participant 4 | 54 | Male | CO | 15 years |
| Participant 5 | 39 | Female | CO | 6 years |
| Participant 6 | 41 | Female | MD | 5 years |
| Participant 7 | 44 | Female | CO | 3 years |
| Participant 8 | 37 | Female | MD** | 11 years |
| Participant 9 | 41 | Female | MD | 8 years |
| Total (n) = 14 | | | | |

Key

* Assistant Medical Officer

** Medical Officer

*** Clinical Officer

Four themes emerged based on the narratives of CTC managers, DHCP and PLHA with a co-morbid NCD (Fig 1). These included 1) Current situation of services available for care and treatment of NCD among PLHA at CTCs 2) Experiences with access to NCD care and treatment services of PLHA with co-morbid NCDs, 3) Facilitators of Integrated care and treatment for NCDs within CTCs. 4) Perceived barriers to accessing and integrating the care and treatment of NCDs within CTCs.

**Theme 1: Current situation of availability of care and treatment services for NCDs among PLHA in CTCs.** This theme included one category of referrals from CTCs to NCD

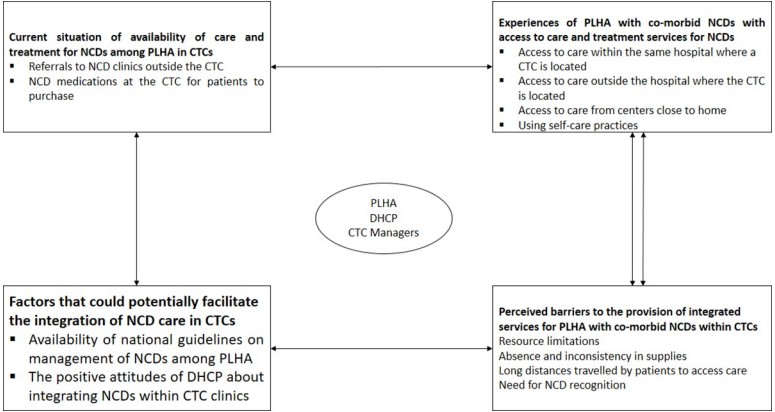

**Fig 1. Summary of findings showing how the four themes relate to each other and the categories within each theme.**

clinics outside the CTC or CTC staff providing prescriptions for NCD medications to be purchased by patients. From the accounts of DHCPs, a substantial number of PLHA with co-morbid NCDs were attended at CTCs, however care and treatment for their NCDs was not provided within the CTCs as narrated by a DHCP in the following quote, *"We receive many patients with hypertension here at the CTC, about 30 patients per week or roughly 7–8 patients a day but we don't provide care for NCDs here* (at the CTC)*" (DHCP F, 41 years)*.

The practices of making patient referrals to NCD clinics outside the CTC or writing prescriptions for NCDs medications at the CTC for patients to purchase was a subtheme indicating how CTCs address the challenge of managing PLHAs with co-morbid NCDs. Both DHCP and CTC managers admitted that treatment services for non-communicable diseases (NCDs) were not available at CTCs, but were offered from separate NCD clinics which are located either within the same hospital or in other public hospitals. At times a prescription is written at the CTC for patients to purchase NCD medications outside the clinic as exemplified by the following quotation; *"Yes. Here* (at CTC) *we screen for NCDs. Once we discover a patient with hypertension, we refer them to the NCD clinic within this hospital or at times at other hospitals, so they attend a different clinic depending on the NCD for their NCD management for instance at the hypertension or the diabetes clinic" (DHCP, F, 25 years)*. The system used in the CTCs to refer patients with NCDs for services was described as strong because it was based on the type and severity of the NCDs PLHA presented with. DHCPs narrated how the referral system works, and I quote:

> *"Yes, our referral system is good. We either refer* (PLHA) *to the specialist or physician depending on their diagnosis and the severity of the NCD. If it happens that they* (PLHA) *come here when the condition (NCD) is already severe we refer them to specialists at the Muhimbili National Hospital.* (DHCP, M, 36 years)

**Theme 2: Experiences with accessing care and treatment services for NCDs of PLHA with co-morbid NCDs.** This theme covered four categories that focus on accessing care for NCDs a) within the same hospital where a CTC is located, b) a hospital outside the hospital where the CTC is located, c) using self-care practices and d) from centers close to home. These forms of access to NCD care were described to be practiced for various reasons, including convenience and to minimize costs; faith as well as religious beliefs were used by PLHA who cannot afford the costs of their NCD care. These participant attitudes towards their NCD care were influenced by their individual experiences and life situations. In this theme the PLHAs shared their experiences of using allopathic or alternative health systems for treatment of their NCDs.

*Accessing care for NCDs within the same hospital where a CTC is located*. The majority of PLHA participants stated that NCD services are offered outside the CTC. Even when the diagnosis of the NCD was made at the CTCs they would be transferred to specific NCD clinics within the facility for their NCD care as demonstrated by the following quote:

> *"At the CTC clinic they said I have to be treated at a different place, not the CTC but at another NCD clinic in the hospital or at any other hospital, because at the CTC they only deal with the ARVs (Patient, F, 52 years)*

*Accessing NCD treatment from a hospital external to the one where the CTC is located.* In some cases, participants accessed NCD services at a facility that was external to the hospital in which their CTC is located and to some PLHA. When this was reported, PLHA found this to be acceptable, mainly because the NCD services they were referred to were offered at no cost

to them. The measures that CTC staff take to seek affordable NCD services for PLHA with NCD comorbidities is illustrated in the following quote:

> "*I received a call from a nurse who works at my CTC in Temeke and she instructed me to go to the Hindu Mandal Hospital* (A private hospital) *where I received treatment for my high blood pressure for free. Until now I haven't paid anything although I am getting good services there. The problem is the distance, but the care is free. That's fair* (Patient, F, 52 years)

Referrals would also be made to public hospitals with specialized services, whereby referral to a secondary level hospital may lead to referral to a tertiary level hospital; requiring several steps in negotiating access to NCD services on the part of the patient as noted in the following quotation:

> "*It was a long time ago when I started the treatment of ARVs at the CTC. I was not feeling okay and felt pain in the shoulder and breast. Then I was advised to go for testing at Mwananyamala and I was referred to Muhimbili after getting the results. Then I was told my heart had expanded and I started attending the clinic there. The doctor wrote laboratory test that I should do (Patient, F, 65 years)*"

*Self-care and attending for NCD care at nearby centers close to home for convenience and to minimize costs*. Patients used several strategies to ensure some access to NCD services given available resources. For example, to minimize transport costs, many patients reported relying on accessing facilities close to home and making decisions themselves about medication use for managing there NCDs. These strategies often circumvent the need for the more expensive close monitoring of treatment progress as elaborated by the following quotations.

> "*In the beginning I used to go to the diabetes clinic that is where I was referred to, at Muhimbili. But then I started realizing that was too far. Because I understand the medication, I usually write them in a book, and I also do self-tests because I have a glucometer. So, if my sugar goes up or if it goes down, I then know what medications and doses to use. Because if it goes down, I drink a soda and it stabilizes, it means the medication is too much. So, I self-test and I treat myself*" (Patient, F, 69 years)

*Faith and religious beliefs used by PLHA who cannot afford the costs for their NCD care*. PLHA with co-morbid NCDs with inability to afford NCDs treatment costs described their dependency on God as a cure for the NCD as narrated in the following example:

> "*I depend on God he is the one who knows my life, I do not have the ability* (money) *to pay for blood pressure drugs. The only support I had was my child who I was depending on to buy these drugs for me and he died. So, it's hard for us to afford even meals, how will I afford drugs*? *I leave that to God. (Patient, F, 70 years)*

**Theme 3: Factors that could potentially facilitate the integration of NCD care in CTCs.** Two categories were included in the third theme. These describe potential enablers of NCD integration like the availability of national guidelines on management of NCDs among PLHA in CTCs and the positive attitudes of DHCP about integrating NCDs within CTC clinics.

*Availability of national guidelines on management of NCDs in PLHA at CTCs*. The DHCP from the CTCs mentioned that they have the national guidelines which help them in providing NCD services accordingly as elaborated in the following quote;

*"We are using the guidelines from the Ministry of Health; everything is explained there on how to manage different NCDs among PLHA.* (DHCP, F, 41 years)

*Positive attitudes of DHCP about integrating NCDs within CTC clinics*. The DHCP and CTC managers felt the need to integrate NCD care within CTC services for reasons based on their experiences with the current lack of integrated care. They noted the advantages to patients in terms of their time, reduced challenges related to navigating their NCD care at different clinics/locations, and reduced out of pocket costs for their NCD care, which unlike ART services is not fully subsidized by the government. These sentiments of providers of CTC managers and DHCP are exemplified by the following quotes:

*"if we can offer NCD services, patients will receive treatment for two conditions in one place. It will also help in reducing costs to patients, if NCD services will be offered for free. You know currently, if the patient does not have money to pay for services provided at the NCD clinic, they are not provided with services for free.* (DHCP, F, 28 years)

*"If NCD and HIV services were to be provided under one roof, it would decrease the stress to the patient of moving from clinic to clinic. Also, it will increase adherence to care because as they move from clinic to clinic, they get tired and even decide not to attend the NCD clinic. At times they* (patients) *have reported to us that after attending their CTC clinic for ART, when they then go to NCD clinics they find the clinic working hours have ended and clinics already closed, thus they miss the NCD services* (on the day of referral)" (DHCP, F, 39 years)

Another CTC manager also explained, *"In fact, it would be better to integrate NCD care into CTC services to reduce the hassle for patients. We get a patient who tells you, "doctor please hurry up, I want to go to my NCD clinic today I have a diabetes clinic* (to attend). *If we were to have diabetes tests here and if we were able to check their blood pressure here, then they would be treated right here and we would be able to monitor their progress, except for those who perhaps have complex conditions that would need referral to specialists* (CTC Manager, F, 51 years)

**Theme 4: Perceived barriers to the provision of integrated services for PLHA with co-morbid NCDs within CTCs.** Eight categories were included in the fourth theme. These were barriers described as potential hindrances to integrating NCD care within CTCs by direct health care providers (DHCP) and CTC managers including; a) resource limitations b) both absence and inconsistency in supplies, c) lack of priority given to recognition and treatment of NCDs in CTCs, d) missing electronic payment systems for NCD care in CTCs e) long distances travelled by patients to access care and f) need for NCD recognition and g) NCD treatment refresher course as well as h) ongoing staff training:

*Resources limitations in terms of medication stocking, data documentation systems and availability of screening tools for NCD care within CTCs.* The hospital systems are not currently designed to facilitate management of NCDs within CTCs. Hospitals do stock NCD medications but do not allocate to CTCs, nor do continuous medical education programmes include regularly training and supporting CTC staff to provide NCD care, as exemplified by the following quotations:

*"There is no medicine for treating NCDs at the CTCs therefore you may find a patient who is living with HIV and also suffering from high blood pressure or diabetes but cannot get service at the CTC for these co-morbidities and must schedule another time to attend the NCD clinic because that's how the hospital is structured,* (CTC Manager, F, 37 years)

*Limited and inconsistent supplies for existing NCD screening instruments in CTCs*. CTC managers noted that though some NCD screening tools were available in their CTCs, there was inconsistent availability of supplies necessary to use these tools. These common challenges, are exemplified by the following quotations, "*Yes, instruments like BP machines, blood sugar testing machines. Yah, we have them. We have weighing scales as well. The biggest challenge is inconsistent supplies for us to be able to use them, for example accessing blood sugar testing strips and machine is a problem here.*, *CTC Manager, M, 50 years)*

*Less priority given to NCD care at CTCs and less awareness of PLHA about NCDs*. The CTC managers felt that patients and DHCP at CTCs expect services available at CTCs to only be the provision of a narrow range of HIV care and ARV treatments for PLHA. Other diseases are not looked upon as a priority by both DHCP and patients as narrated by one DHCP in the following quotation, "*I think NCDs are not given priority here* [at CTC] *and even the patients we treat do not give priority to their NCDs care they tend to demand more of the ARVs refill services than care for their NCDs. It is simply because the CTCs do not prioritize such diseases and there is nothing special, we (DHCP) are doing to ensure PLHA understand and demand for NCDs services here. Even if screening services for hypertension and random glucose are available it is easy for PLHA to just collect their ARVs without noticing that such services exist. (DHCP, M, 36 years)*

This DHCP quotes, is supported by PLHA descriptions of their challenges of not knowing about their NCDs such as hypertension, how it is treated as well as how to access NCD care. When medications are given to them, they may not have clarity about what medication should be used for what condition. This is exemplified by these concerned patient's quotes:

*"Mmhm! Firstly, for us PLHA with high blood pressure, we do not understand the blood pressure medications, they (DHCP) mix our medications and do not tell us what medication treats what, they just prescribe them silently and they (DHCP) do not help to make us understand better we just take them. At times friends will tell us those don't work, use this other one. You see and we are not sure what to do. We need them to tell us more. (Patient, F, 60 years)*

*Lack of regular training of CTC HCP on NCDs management*. A DHCP mentioned that, it's a long time since they received NCD management training, and when they did, it was a short training course. She felt a need for refresher training courses for providers as noted in the following quotation:

*"I manage NCD patients using the knowledge I got from college. As I told you, I am an assistant medical officer so I was taught in college. . . this is not enough; it is good we get updates because it has been a while since I went to school and treatments have changed. (HCP, F, 44 years)*

*Treatment costs for NCDs may be prohibitive*. The DHCPs noted that costs for NCD care are currently a major challenge for many PLHA with co-morbid NCDs. They expressed a concern that currently, at the NCD clinics, diagnosis and treatment is expensive; in addition to a registration fee, patients pay for laboratory and other investigation costs, as well as for treatment. Attending NCD care may not, hence be sustainable. This is narrated as follows by a health care worker:

*"In the NCD clinic, treatment is expensive. The PLHA receives consultation service and ARVs for free, but for NCD care they must pay for laboratory tests or any other test requested, whether it's a full blood picture, or chest x-ray they must pay including for purchasing drugs.*

*So, treatment is expensive and PLHA can't afford it. As a result, some PLHA abandon the care. (Participant, HCP, F, 28 years)*

Moreover, a PLHA added and I quote;

"*(sighs) no. we are not satisfied with the costs at all. We have no ability to buy the medication. Being a street cook and I have to try and save and save and save until I get to enough for one bottle which is 17,000–18,000* (shillings); *and I need to be injected with two [bottles]! So sometimes I have no choice but to skip using medication. . . (Patient, F, 39 years)*

*An integrated payment system for NCD care in CTCs might not be acceptable.* HCPs noted that PLHA are used to receiving the fully government subsidized services offered at CTCs. If NCD care and costs were to be integrated in the CTC services, patients may not accept it because they have become accustomed to services at no cost to them. The following quotations exemplify HCPs concerns:

"*Integration is possible but the biggest challenge is on the side of finances because in that clinic* (NCD) *the PLHA is supposed to have ten thousand* (shillings) *for registration and this is not inclusive of medication. Still the patient will need to pay more for NCD treatment and you may find that the patient does not have the money. If that system is introduced here at CTCs patients will not agree because they are used to free services (Participant, DHCP, 28 years)*

*Long distances from home to NCD clinics.* Some PLHA were attending NCD clinics located long distances from their homes, this meant waking up very early in the morning in order to attend and incurring high transport costs as described in the following quotations:

"*. . .until I arrive at the clinic, I need to get out of the house at 04:00 am and arrive at the clinic at 06:00 am to be there early enough to get treated and leave on time. This is really far (Patient, F, 52 years)*

## Discussion

Key findings of narrative data from all participants are presented in four themes including; the current situation of services available for care and treatment of NCDs among PLHA in CTCs; experiences of PLHA with co-morbid NCDs in accessing services for NCDs; potential facilitators for NCDs integrated services within CTCs and perceived barriers for accessing and integrating NCD services within CTCs. Most participants noted the limited presence of NCD integrated services within HIV CTCs, and all expressed a need for integrating diagnosis and treatment of NCDs within CTCs. Some authors have outlined the importance of bundling HIV and NCD services within HIV CTC including having adequate treatment adherence supports for improved quality of life, close monitoring of biomarkers that may be predictive of the development of NCDs and provision of psychosocial supports Haregu et al. (2015) [7]. The WHO, through its health system straightening programme, also highlights a number of benefits associated with integrated NCD services in HIV care and treatment service platforms, including improved health and longevity and reductions in the NCDs burden through early screening and treatment [14].

Our findings show PLHA with co-morbid NCDs, experience the current model of access to NCD services as occurring outside the HIV services platform. This approach may risk poor

continuity of care and patient progress, as well as patient time constraints when navigating between and within the NCD and HIV service platforms, especially in facilities that do not house both these service platforms. Most patient participant narratives indicated a need for NCD treatments within CTC, and awareness of the advantages of such integration for their health and wellbeing. Studies show that if NCDs are diagnosed early, there is a greater opportunity for reducing NCD associated complications [15]. This may only be possible if resources at CTCs allow for routine NCD screening, diagnosis, treatments and related documentation. According to the UNAIDS Second Meeting of the WHO Global Coordination Mechanism, integrated NCD services on HIV service platforms can result into better screening, treatment and cure for NCDs [14]. The Tanzania Strategic and Action Plan for the Prevention and Control of NCD 2016–2020 has highlighted the importance of shifting from curative to prevention and control measure [16]. This can only be achieved through clearer guidelines for comprehensive routine screening, diagnosis and early treatment on relevant health services platforms.

A number of factors that may facilitate integration of NCD services with in CTC emerged from the narrative data, ranging from the availability of national guidelines for the management of NCDs among PLHA to positive attitudes among PLHA and DHCP towards integrated services. The availability of guidelines for the management of major NCDs in in primary health care facilities in African countries is low and only reported at 17% [17]. This is similar to healthcare's narrations in the interviews where they acknowledged the availability of guidelines for the management of major NCD such as hypertension and diabetes, while routine screening being behind. The positive attitudes among PLHA and DHCP participants towards integration of NCD services within CTC were in part a result of the perceived advantages in terms of perceived reduction in PLHA time when utilizing health services and the potential for them reducing out-of-pocket health care costs.

Among the reported factors that may hinder integration of NCD within CTC were: Resource limitations in terms of medication stocking for treatment of NCDs, access to NCD data documentation systems and availability of screening tools and point of care aids for NCD care within CTCs. Similar findings on limited or no NCD medical products or supplies in HIV health care services have been reported by Lamptey P., et al. [18]. Both the observation checklist and healthcare provider accounts suggested either limited or inconsistent supplies for screening of existing NCDs in CTCs. This was also reported by Mpondo et al. [19], where authors questioned the level of preparedness for NCD diagnosis and treatment in lower level public hospitals. We also found that continuing medical education offered to CTC staff did not include screening, diagnosis and management of co-morbid NCDs, reported by DHCP informants. Poor training of staffs for NCD management has also been reported by other research teams in Tanzania [9, 19].

Other structural factors such as treatment costs and hospital payment systems may also hamper integration of NCD care within CTC services. Our findings show that while |CTC services for PLHA are currently provided at no costs to patients (i.e., fully subsidized by the government), facilitating access to and affordability of services to the user. However, this is not the case for NCD services, and the Tanzania Strategic and Action Plan for Prevention of Non-Communicable Diseases (NCD) 2016–2020, highlights lack of funding as a major implementation challenge [16]. The absence of fully subsidized NCD services, is evidenced by the existence of a reported cost-share system, where patients are expected to pay out-off pocket expenses as contributions for some NCD services they receive, for example medications and laboratory investigations. The Global Action Plan for Prevention and Control of NCDs 2013–2020 stipulates a need for support at a national level to enable access to NCD drugs, as an approach for the prevention of heart attacks and stroke [20]. Future studies in Tanzania may need to explore more cost-effective ways of purchasing NCD drugs (e.g. bulk purchase) [21], and conducting

routine NCD relevant laboratory investigations (e.g. use of high through put laboratory equipment) [16].

## Study strengths

This study provides an overview of the experiences of PLHA, healthcare providers and CTC managers and direct health care providers in the integration of NCD services within the HIV treatment services platform and the barriers and potential facilitators of such integration. Inductive content analysis of transcripts with the support of illustrative quotations from study participants provide a better understanding of the current integration status and barriers to implementation that perhaps need to be addressed. Analysis of narrative information derived from three different types of informant strengthens the validity of the study findings [13].

## Study limitations

The use of the data from the five large CTC in Dar es Salaam may not have been a representative of the all PLHA, as CTCs with lower patient volume, may have used alternative strategies to support PLHA with NCD co-morbidity. However, the findings do provide a general overview on the experiences of the majority of PLHA with NCD co-morbidities, their attitudes towards integrating HIV and NCD services, and perceived barriers to such integration of services platform. Conducting IDs interviews at the clinic settings, where the participants receive their care/services may have also influenced them towards giving socially acceptable answers to the interview questions. The researchers mitigated this bias by asking open-ended questions, so that to make participants respond in a way that is not socially desirable.

## Conclusion

There was a positive attitude among PLHA and healthcare workers towards integration of NCD services within CTC. This was attributed by the perceived benefits associated with the integration of NCD services under one clinic, i.e. NCD services for individuals who are known to have NCD prefers to receive the care with in CTC where they receive their ART services, however the series were no rooved within the CTC. Due to financial challenges PLHA to access transport, they have decided to receive their services at the nearby health care facilities. However, there were number of factors which may impede NCD integration care and services including; limited and inconsistent supplies such as screening equipment, medications; insufficient awareness of NCD within PLHA; lack of adequate training to healthcare workers on management of NCD and treatment costs and payment system.

## Recommendations

Integration of NCD care and services with in CTC is very important approach in order to curb the morbidity and mortality related to NCD complications. Government and stakeholders jointly deal with the challenges which hinders integration of NCD care and services within CTC so that screening and management of NCDs among PLHA's opportunity are not missed.

## Supporting information

**S1 File. Semi-structured interview guide for PLHA.**
(DOCX)

**S2 File. Semi-structured interview guide DHCP & CTC managers.**
(DOCX)

**S3 File. Observational checklists at CTC on NCD issues.**
(ZIP)

**S4 File. Transcripts PLHA.**
(ZIP)

## Acknowledgments

The authors would like to thank the participants, that is, PLHA, direct healthcare providers and CTC managers for their willingness to participate in the study. I would also like to acknowledge Prof. Sylvia Kaaya from Muhimbili University of Health and Allied Science for her close supervision of this research project.

## Author Contributions

**Formal analysis:** Tausi Haruna, Magreat Somba.

**Funding acquisition:** Hellen Siril.

**Methodology:** Tausi Haruna, Magreat Somba, Gladys Mahiti, Anna Minja.

**Project administration:** Ferdinand Mugusi.

**Software:** Hellen Siril.

**Supervision:** Hellen Siril, Gladys Mahiti, David Urassa, Edith Tarimo, Ferdinand Mugusi.

**Writing – original draft:** Tausi Haruna.

**Writing – review & editing:** Hellen Siril, Francis August, David Urassa, Edith Tarimo, Ferdinand Mugusi.

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
