## [Decision Letter · Decision Letter 0]

8 Mar 2021

PONE-D-21-03870

Factors Hindering Integration of Care for Non-Communicable Diseases within HIV care services in Dar es Salaam, Tanzania; The Perspectives of Health workers and People Living with HIV

PLOS ONE

Dear Dr. Haruna,

Thank you for submitting your manuscript to PLOS ONE. After careful consideration, we feel that it has merit but does not fully meet PLOS ONE’s publication criteria as it currently stands. Therefore, we invite you to submit a revised version of the manuscript that addresses the points raised during the review process.

We look forward to receiving your revised manuscript.

Kind regards,

Tareq Mukattash

Academic Editor

PLOS ONE

Journal Requirements:

2. When reporting the results of qualitative research, we suggest consulting the COREQ guidelines: http://intqhc.oxfordjournals.org/content/19/6/349. In this case, please consider including more information on the number of interviewers, their training and characteristics; and please provide the interview guide used.

In the Methods section, please provide additional information regarding the interview guide development process, including the theories or frameworks which were employed.

Please include in your Methods section (or in Supplementary Information files) the participating hospitals/institutions.

Please provide the observational checklist as Supporting File.

3. We note you have included a table to which you do not refer in the text of your manuscript. Please ensure that you refer to Table 1 and 2 in your text; if accepted, production will need this reference to link the reader to the Table.

Reviewers' comments:

Reviewer's Responses to Questions

**Comments to the Author**

1. Is the manuscript technically sound, and do the data support the conclusions?

Reviewer #1: No

Reviewer #2: Yes

2. Has the statistical analysis been performed appropriately and rigorously? 

Reviewer #1: N/A

Reviewer #2: N/A

3. Have the authors made all data underlying the findings in their manuscript fully available?

Reviewer #1: No

Reviewer #2: No

4. Is the manuscript presented in an intelligible fashion and written in standard English?

Reviewer #1: No

Reviewer #2: Yes

5. Review Comments to the Author

Reviewer #1: Thanks for considering PLOS ONE for your submission.

I have noted points that I would like to share about your submitted work:

1. The submission contains many writing mistakes (typos, sentence structure issues e.g. run on sentences) (I recommend using english writing aid tools), consistency would help (checklist was written in many different ways)

2. The short title is still long I recommend making it shorter

3. Key words need to be revised and modified to facilitate the chance of locating your work if it gets published

4. I do not agree with including too many subheadings/subtitles as you have done with every section in introduction, try to provide coherent paragraphs , instead.

5. Providing numbers for lines would have made my job and yours easier to track my comments.

6. The second sentence in third paragraph in the introduction needs a reference.

7. The ART abbreviation was not defined at first mention

8. Regarding study design, I did not find the primary assumptions from authors. I think this would raise the issue of personal bias while producing useful data from study participants, which may lead to biased hypothesis representing the phenomenon of interest.

9. I think some of your eligibility and exclusion criteria are redundant.

10. Both sections of sampling approach and study settings include same information (redundant)

11. I recommend that you include further details about the development of the guide of semistructured interviews.

12. Data analysis section includes some of the results , and it lacks the inductive content analysis process of analysis. More details describing the analysis is needed to show the link between extracted data and your reported results.

13. I recommend that you include further details about the open coding/grouping steps and category creation and abstraction processes.

14. I would like to see better organization by providing a conceptual model with your identified categories to give a meaningful sense of describing the selected phenomenon to increase readers understanding and help in generating knowledge.

15. I would suggest to provide names or headings of higher order categories given your driven content characteristic words.

16. Providing Tables/figures including well labeled sub-categories and generic categories and main categories would be helpful

Thank you

Reviewer #2: This is an interesting qualitative study that aimed to explore the perceptions of HIV patients and healthcare workers on facilitators or barriers of the recognition and integration of care for non-communicable diseases within treatment clinics.

Overall, the paper is well written however, it a bit lengthy and both the introduction and the methods can be reduced.

- I enjoyed how the authors narrated the quotes in the results, however, I cannot see the benefit from table 3. It would be easier to limit the examples on themes in one place. I prefer to keep them in the results section and take out table 3.

- i recommend to move Onsite checklist-based observation findings to the prior to the results of thematic analysis.

in table 1: I am not sure if you are meant to say "hypertension" instead of "hypotension"

in table 1: please check the (%) they do not add up for example:

Male 4(14.8%)

Female 23(85.1%)

total is not 100%

same for

Business 16(59.2%)

Employed 1(3.7%)

Unemployed 10(37.0%)

in table 1: ulcers! what kind of ulcers?

I am not sure if table 4 is needed, the summarized results in the Onsite checklist-based observation findings in the results section can be sufficient.

Limitations: there are more limitations to be discussed related to the use of qualitative analyses.

there are few grammatical errors such as " what facilitates or hinder"

6. PLOS authors have the option to publish the peer review history of their article (what does this mean?). If published, this will include your full peer review and any attached files.

Reviewer #1: No

Reviewer #2: No

---

## [Author Response · Author response to Decision Letter 0]

27 May 2021

Journal Requirements: 1. Please ensure that your manuscript meets PLOS ONE's style requirements, including those for file naming. The PLOS ONE style templates can be found at

 I have visited the link for guidance and now files meet the requirements of the PLOS ONE’s style 

When reporting the results of qualitative research, we suggest consulting the COREQ guidelines: http://intqhc.oxfordjournals.org/content/19/6/349. In this case, please consider including more information on the number of interviewers, their training and characteristics; and please provide the interview guide used. 

Research assistants were trained. The research assistants had a mix of pre-graduate diplomas in health and social science para professional field (n=3) postgraduate degree in Public health (n=1) and undergraduate degrees in the social sciences or Law (n=2) fields. All RAs also were up to date on research ethics certification (FHI360 certification course). Find attached interview guides in the supplementary files. See Page 7.

In the Methods section, please provide additional information regarding the interview guide development process, including the theories or frameworks which were employed. 

Information regarding development of interview guide has been added including framework. See Page 5-6

Please include in your Methods section (or in Supplementary Information files) the participating hospitals/institutions. 

We have added…five major CTC namely,

Mwananyamala

Temeke Regional Referral Hospital 

Sinza Health Centre

Amana regional Referral Hospital

Vijibweni health Centre

Because this was a small study, we have decided not to include names of hospitals in the manuscript to protect DHCP and CTC managers but I have included permission letters for data collection in the Supplementary files

Observational checklist at CTC on NCD issues

Please provide the observational checklist as Supporting File. Attached as Supporting Files

We note you have included a table to which you do not refer in the text of your manuscript. Please ensure that you refer to Table 1 and 2 in your text; if accepted, production will need this reference to link the reader to the Table. Thank you 

We have indicated the Tables in the document (Table 1 is for example of the analysis process of the main theme, categories and quotes, Table 2 Demographics data for PHLA and Table 3 Socio-demographics data for DHCP and CTC managers. See Page 8,10 & 11.

Reviewer's Responses to Questions

Comments to the Author 

1. Is the manuscript technically sound, and do the data support the conclusions?

Reviewer #1: No 

Reviewer #2: Yes 

Thank you for our comment, we have now improved more on conclusion to make it appropriately based on the data presented in the manuscript. See page 24

(Thank you for the comment)

2. Has the statistical analysis been performed appropriately and rigorously? 

Reviewer #1: N/A

Reviewer #2: N/A 

3. Have the authors made all data underlying the findings in their manuscript fully available?

Reviewer #1: No

Reviewer #2: No 

All interviews transcripts and checklists have been provided in the supplementary files.

4. Is the manuscript presented in an intelligible fashion and written in standard English?

Reviewer #1: No 

Reviewer #2: Yes 

Thank you for the comment, the manuscript now has been reviewed with professional language reviewer and grammatical errors have been checked and corrected) See page 1 to the end

(Thank you for the compliment)

Reviewer #1 Reviewer #1: 

Thanks for considering PLOS ONE for your submission.

I have noted points that I would like to share about your submitted work:

1. The submission contains many writing mistakes (typos, sentence structure issues e.g. run on sentences) (I recommend using English writing aid tools), consistency would help (checklist was written in many different ways). 

Thank you for your comment, we have now sent the work to language reviewer and now the typos, sentence structures and use of English aid writing tools have been used to support writing) See page 1 to the end (1-27)

2.The short title is still long I recommend making it shorter. Thank you for the comment, we have now made a shorter title and make it shorter. Now it reads as" Factors Hindering Integration of Care for Non-Communicable Diseases within CTCs"

3. Key words need to be revised and modified to facilitate the chance of locating your work if it gets published Thank you for the comment. We have now modified and revised the key words and now it reads Integration, Non-Communicable Disease, Qualitative, Tanzania, PLHA, Clinic and Treatment Centre, Hypertension, Diabetes.

4. I do not agree with including too many subheadings/subtitles as you have done with every section in introduction, try to provide coherent paragraphs, instead. 

Thank you for the comment we have now reduced the subtitles in every section and provided coherent paragraphs. The retained subtitles are for helping a reader to follow through See pages 1 to end (1-27)

5. Providing numbers for lines would have made my job and yours easier to track my comments.

(Thank you for the comment, we have now inserted line numbers to make our job easier. See page 1 to the end)

6. The second sentence in third paragraph in the introduction needs a reference 

Reference inserted. See Page 3-4

7. The ART abbreviation was not defined at first mention 

Thank you for the comment, we have now defined the ART abbreviation when first mentioned. See page 4)

8. Regarding study design, I did not find the primary assumptions from authors. I think this would raise the issue of personal bias while producing useful data from study participants, which may lead to biased hypothesis representing the phenomenon of interest. 

Thank you for the comment, we have now added a sentence…We prepared In-depth interview guides based on current literature on the recognition and integration of care for NCDs within CTC and trustworthiness section as an assumption to show authenticity of the design and quality of data. See page 10

9. I think some of your eligibility and exclusion criteria are redundant. Removed…in the targeted 5 district referral hospital

Removed....and facility/ CTC heads also known as CTC in charge.

Removed… who were not receiving ART

Deleted or did not provide signed informed consent as well as PLHA newly enrolled to 

Deleted. those not directly involved with caring for PLHA and newly employed

We have merged this section into Study setting as per COREQ checklist See Page 6.

10. Both sections of sampling approach and study settings include same information (redundant). Thank you for the comment we have now removed sampling approach section and we have recasted study settings section to be more focus and avoid redundancy. See Page 6-7.

11. I recommend that you include further details about the development of the guide of semi structured interviews. Thank you for the comment, we have now added section data collection tools and we have included information on the development of the interview guide. Refer page 5-6

12. Data analysis section includes some of the results, and it lacks the inductive content analysis process of analysis. More details describing the analysis is needed to show the link between extracted data and your reported results. Thank you for the comment, we have now recasted the data analysis section and put more description of analysis showing the link between the extracted data analysis and reported results) See page 8

13. I recommend that you include further details about the open coding/grouping steps and category creation and abstraction processes (Thank you for the comment, we have now recasted the data analysis section and put more description of analysis showing the link between the extracted data and analysis and reported results and put example of analysis See Table 1) See page 8 where the changes of analysis section has been done

14. I would like to see better organisation by providing a conceptual model with your identified categories to give a meaningful sense of describing the selected phenomenon to increase readers understanding and help in generating knowledge. 

Conceptual model has been included. See uploaded figure 1

15. I would suggest to provide names or headings of higher order categories given your driven content characteristic words. (We have decided to provide themes which is more of abstraction level from the analysis we therefore suggest for maintaining those headings) See page 12-20.

16. Providing Tables/figures including well labeled sub-categories and generic categories and main categories would be helpful.

We have formulated a table showing some of the theme and categories (Refer Table 1) and figure of summary of results. See page 8 & 12

Reviewer #2 Overall, the paper is well written however, it a bit lengthy and both the introduction and the methods can be reduced. We have reduced some of the repeating quotes and merged some of the subtitle

We now have 27 pages from 35 pages 

I enjoyed how the authors narrated the quotes in the results, however, I cannot see the benefit from table 3. It would be easier to limit the examples on themes in one place. I prefer to keep them in the results section and take out table 

We have removed Table 3 in the results section. See page 9

i recommend to move Onsite checklist-based observation findings to the prior to the results of thematic analysis. 

Thank you for your comment

We have now moved Onsite checklist-based observation findings before IDIs results. See Page 9 & 10

in table 1: I am not sure if you are meant to say "hypertension" instead of "hypotension"

The word hypotension has been changed to hypertension. Please check Table 2. See page 10

in table 1: please check the (%) they do not add up for example: 

Male 4(14.8%)

Female 23(85.1%)

total is not 100%

same for

Business 16(59.2%)

Employed 1(3.7%)

Unemployed 10(37.0%) Checked and corrected

Male 4(14.8%)

Female 23(85.2%)

Business 16(59.3%)

Employed 1(3.7%)

Unemployed 10(37.0%)

See Page 10

in table 1: ulcers! what kind of ulcers? Ulcers represent all types of ulcers See page 10

I am not sure if table 4 is needed, the summarized results in the Onsite checklist-based observation findings in the results section can be sufficient. 

We have removed table 4 (Refer page 26 & 27)

Limitations: there are more limitations to be discussed related to the use of qualitative analyses. 

Thank you for this comment

We have now added another limitation related to study setting.. Conducting IDs interviews at the clinic settings, where the participants receive their care/services may have also influenced them towards giving socially acceptable answers to the interview questions. The researchers mitigated this bias by asking open-ended questions, so that to make participants respond in a way that is not socially desirable. See page 23 & 24

there are few grammatical errors such as " what facilitates or hinder" We have checked for grammatical errors See page 1 to end (27).

---

## [Decision Letter · Decision Letter 1]

28 Jun 2021

Factors Hindering Integration of Care for Non-Communicable Diseases within HIV care services in Dar es Salaam, Tanzania; The Perspectives of Health workers and People Living with HIV

PONE-D-21-03870R1

Dear Dr. Haruna,

We’re pleased to inform you that your manuscript has been judged scientifically suitable for publication and will be formally accepted for publication once it meets all outstanding technical requirements.

Kind regards,

Tareq Mukattash

Academic Editor

PLOS ONE

Additional Editor Comments (optional):

Reviewers' comments:

Reviewer's Responses to Questions

**Comments to the Author**

1. If the authors have adequately addressed your comments raised in a previous round of review and you feel that this manuscript is now acceptable for publication, you may indicate that here to bypass the “Comments to the Author” section, enter your conflict of interest statement in the “Confidential to Editor” section, and submit your "Accept" recommendation.

Reviewer #2: All comments have been addressed

2. Is the manuscript technically sound, and do the data support the conclusions?

Reviewer #2: Yes

3. Has the statistical analysis been performed appropriately and rigorously? 

Reviewer #2: Yes

4. Have the authors made all data underlying the findings in their manuscript fully available?

Reviewer #2: Yes

5. Is the manuscript presented in an intelligible fashion and written in standard English?

Reviewer #2: Yes

6. Review Comments to the Author

Reviewer #2: Thank you for addressing all of my comments. I am satisfied with the revised manuscript. I have no further comments.

7. PLOS authors have the option to publish the peer review history of their article (what does this mean?). If published, this will include your full peer review and any attached files.

Reviewer #2: No

---

## [Editor Report · Acceptance letter]

4 Aug 2021

PONE-D-21-03870R1 

Factors Hindering Integration of Care for Non-Communicable Diseases within HIV care services in Dar es Salaam, Tanzania; The Perspectives of Health workers and People Living with HIV 

Dear Dr. Haruna:

I'm pleased to inform you that your manuscript has been deemed suitable for publication in PLOS ONE. Congratulations! Your manuscript is now with our production department. 

Kind regards, 

on behalf of

Dr. Tareq Mukattash 

Academic Editor

PLOS ONE